# A Novel Exercise Facilitation Method in Combination with Cognitive Behavioral Therapy Using the Ikiiki Rehabilitation Notebook for Intractable Chronic Pain: Technical Report and 22 Cases

**DOI:** 10.3390/healthcare9091209

**Published:** 2021-09-14

**Authors:** Shinji Kimura, Masako Hosoi, Naofumi Otsuru, Madoka Iwasaki, Takako Matsubara, Yasuyuki Mizuno, Makoto Nishihara, Takanori Murakami, Ryo Yamazaki, Hajime Ijiro, Kozo Anno, Kei Watanabe, Takuya Kitamura, Shouhei Yamada

**Affiliations:** 1Department of Rehabilitation Medicine, Niigata University Medical and Dental Hospital, Niigata 951-8520, Japan; madoka.iwasaki12@gmail.com (M.I.); ryosanjin@gmail.com (R.Y.); hjij0227@yahoo.co.jp (H.I.); hawaiririuokarani@yahoo.co.jp (S.Y.); 2Department of Psychosomatic Medicine, Kyushu University Hospital, Fukuoka 812-8582, Japan; hosoi.masako.642@m.kyushu-u.ac.jp (M.H.); anno.kozo.379@m.kyushu-u.ac.jp (K.A.); 3Institute for Human Movement and Medical Sciences, Niigata University of Health and Welfare, Niigata 950-3198, Japan; otsuru@nuhw.ac.jp; 4Department of Physical Therapy, Faculty of Rehabilitation, Kobe Gakuin University, Kobe 651-2180, Japan; matsubar@reha.kobegakuin.ac.jp; 5Department of Psychosomatic and General Internal Medicine, Kansai Medical University, Hirakata 573-1010, Japan; mizunoy@hirakata.kmu.ac.jp; 6Multidisciplinary Pain Center, Aichi Medical University, Nagakute 480-1195, Japan; ck410621@gmail.com; 7Department of Rehabilitation Medicine, Sapporo Medical University, Sapporo 060-8556, Japan; takanori@sapmed.ac.jp; 8Department of Orthopedic Surgery, Niigata University Medical and Dental Hospital, Niigata 951-8520, Japan; watakei1502@med.niigata-u.ac.jp; 9Department of Physical Therapy Course, Niigata University of Rehabilitation, Murakami 958-0053, Japan; kitataku11260207@gmail.com

**Keywords:** psychological therapy, education, notebook, catastrophizing thoughts, quality of life

## Abstract

Recent clinical practice guidelines for chronic pain indicate, with a high evidence level, that the combination of exercise and cognitive behavioral therapy (CBT) is effective. The purpose of this study was to evaluate the effectiveness of an exercise facilitation method in combination with CBT using the “Ikiiki Rehabilitation Notebook” for patients with intractable chronic pain. “Ikiiki” means active in Japanese. A total of 22 cases with chronic low back (*n* = 13), lower extremity (*n* = 8), or neck (*n* = 1) pain were treated using this notebook. Two cases dropped out, leaving 22 cases. Each case was evaluated in terms of the numerical rating scale (NRS) of the pain, activities of daily living (ADL), pain catastrophizing scale (PCS), and quality of life (QOL) at pretreatment and post-treatment. The endpoint of the method was to achieve the long-term goals set by the patients. The mean treatment period was 11.2 months. The outcomes were as follows: improvement of presenteeism: nine cases; enhanced participation in hobbies: seven cases; improved school attendance: two cases; return to work: one case; improvement of self-care and/or self-efficacy: three cases. The NRS, ADL, PCS, and QOL were significantly improved after the treatment. This method is possibly valuable for educating patients about the cause and treatment of chronic pain and actively facilitating exercise and social participation. Further studies are needed to investigate the effectiveness of using this notebook for the patient with intractable chronic pain.

## 1. Introduction

An epidemiologic survey of approximately 11,000 Japanese people aged 18 years and over revealed that the prevalence of chronic pain (pain score of >5 on a Visual Analogue Scale lasting at least 6 months) was 15.4% [1]. Similarly, in a large-scale survey, 17 million reported chronic pain in 2004, with that number increasing to 23.15 million in 2010 [2], suggesting that the number of patients with chronic pain is growing, especially with rapid aging in Japan. According to a satisfaction survey of chronic pain treatment in Japan, 64% of patients are not satisfied and 49% of patients have changed their medical facilities to seek relief [1]. On this basis, in Japan, the degree of satisfaction with chronic pain treatment is low; approximately half of the patients are doctor shopping. Those with complaints of chronic pain have high unemployment and dropout rates, and the large socio-economic loss is a serious social problem [1].

For high-quality guidelines, exercise therapy for chronic low back pain is recommended under the supervision of a physical therapist [3]. In addition to reducing pain, exercise therapy improves muscle strength, lumbar range of motion, and quality of life (QOL) [4]. Japanese guidelines for the treatment of chronic pain published in 2018 [5] report the usefulness of exercise therapy and cognitive behavioral therapy (CBT). For Clinical Question (CQ) 44 in the guidelines, “Is the introduction of CBT and education into rehabilitation and its application to treatment effective in treating chronic pain?”, both CBT and patient education scored a grade of 1B (1: strongly recommended, B: the estimated value of an effect is moderately reliable). The combination of CBT and therapeutic exercise was found to be more effective for improving pain and physical function than general physiotherapy [6]. Therefore, we developed the “Ikiiki Rehabilitation Notebook” based on the concept of CQ 44 in the Japanese guidelines for the treatment of chronic pain [7].

The purpose of this study was to describe a novel exercise facilitation method combined with CBT using the “Ikiiki Rehabilitation Notebook” for patients with intractable chronic pain, and to evaluate the effectiveness of using this method for patients.

## 2. Materials and Methods

### 2.1. Patient Information

The patients in the present study were 8 men and 16 women (*n* = 24) ranging in age from 19 to 77 years (average ± SD: 51 ± 18). Chronic pain locations were low back (*n* = 14), lower extremities (*n* = 8), or neck (*n* = 2). Clinical diagnoses included chronic low back pain (*n* = 10), lumbar spondylosis (*n* = 5), chronic lower extremity pain (*n* = 3), chronic knee pain (*n* = 2), chronic neck pain (*n* = 2), osteoarthritis of the knee (*n* = 1), and chronic plantar pain (*n* = 1). The period (months) from the occurrence of pain to the first visit to the Niigata University Hospital ranged from 5 to 200 months (average ± SD: 58.0 ± 43.5). The duration of the treatment using this notebook was 10.8 ± 4.7 months on average, ranging from 3 to 22 months. The average number of medical facilities visited by the patients before coming to the Niigata University Hospital was 3.8 ± 2.0.

Inclusion criteria in this study were as follows: (1) Numerical Rating Scale (NRS) for pain > 3/10, (2) all patients had pain somewhere in the body, other than headaches, for more than 3 months, resulting in disability in activities of daily living (ADL), (3) inadequate effects of medication to effectively improve the pain intensity or ADL, and (4) patients were eager to do rehabilitation therapy. Exclusion criteria were as follows: (1) patients were under litigation or had accepted insurance payments for a traffic accident and (2) patients suffered from severe psychological diseases, such as depression and schizophrenia.

### 2.2. Ethical Consideration

The present study received ethical approval from the Ethics Committee of Niigata University (No.2016-0090), and written informed consent was obtained from each subject prior to this study.

### 2.3. Outcome Measures

The patients were evaluated at pretreatment, every 3 months, and at post-treatment, as described previously [8]. The NRS was used to evaluate the pain intensity from 0 to 10 [9]. The Pain Disability Assessment Scale (PDAS) is a questionnaire for assessing the influence of pain on a patient’s daily life, including the degree of physical activity and mobility [10], with patients assigning a score of 0 to 3 points for ADL, social life activity, etc. The higher the score, the more severe the dysfunction. The cut-off score is 10 points. The Hospital Anxiety and Depression Scale (HADS) is used to evaluate anxiety and depression [11] without consideration of physical symptoms. It consists of 14 items, 7 for anxiety and 7 for depression. The higher the score, the stronger the anxiety and/or depression. The Pain Catastrophizing Scale (PCS) is used to evaluate pain catastrophizing [12], which consists of 3 sub-items: rumination, i.e., thinking repeatedly about pain; feelings of helplessness regarding the pain; and magnification, i.e., the patients experience pain disproportionate to the stimulus. The PCS comprises 13 questions, and the higher the score, the stronger the catastrophic thinking. The cutoff score is 30 points. The Athens Insomnia Scale (AIS) is used to assess insomnia [13], which includes 8 questions. A score of 4 to 5 indicates suspected insomnia, and the score of 6 points or more means probable insomnia. The Pain Self-Efficacy Questionnaire (PSEQ) is used to evaluate self-efficacy, i.e., self-confidence, while in pain [14]. The PSEQ comprises 10 questions with response values ranging from 0 to 6 points. The higher the score, the higher the self-efficacy. Fewer than 20 points is considered to indicate low self-efficacy. The EuroQol 5 Dimension (EQ-5D) is used to evaluate the QOL [15]. Patients select 1 of 3 levels (no problem, some problems, severe problems) in 5 dimensions: mobility, self-care, usual activities, pain/discomfort and anxiety/depression. The score is assigned a numerical value from 0 to 1.0 using a conversion table.

To clarify the pretreatment factors affecting the QOL at post-treatment, we examined the relationship between all data at pretreatment and the EQ-5D at post-treatment.

### 2.4. Background and Development of the Ikiiki Rehabilitation Notebook

In a study reported by Hirase et al., to evaluate a combined protocol of CBT and exercise therapy, physical therapists providing support for the community exercise classes checked the participant’s diaries once a week and advised participants to alter their pain awareness by focusing on increasing their daily activities [16]. The diary used in this study, however, did not include queries of psychosomatic content such as “thoughts” and “feelings”, which is important to relieve the catastrophizing thoughts that are characteristic of chronic pain patients. To the best of our knowledge, there are no reports of studies in which participants used a notebook to record their thoughts, feelings, and messages to thank oneself for the effort toward combined CBT and exercise therapy [8,17].

Therefore, the first author of the present study (a rehabilitation physician and board member of the Japanese Association for the study of Musculoskeletal Pain) assumed leadership of the study and recruited members from the boards of the same associations to develop the notebook used here. Members comprised 2 psychosomatic physicians, a psychiatrist, a physical therapist, 2 rehabilitation physicians, and a pain clinician. For the exercise therapy, the physical therapist, rehabilitation physicians, and pain clinician mainly discussed examples of short-term and long-term goals, exercise therapy types and methods, and methods for monitoring the number of steps taken per day. For CBT, the psychosomatic physicians and psychiatrist discussed methods of describing thoughts and feelings in daily records based on CBT theory. The idea of writing a “message to thank oneself for the effort”, as the most distinctive feature of this notebook, was suggested by the psychosomatic physician (second author in the present study) from the perspective of improving the low self-efficacy that is considered a characteristic of chronic pain patients. Following submission of the springboard for discussion from each member, all members together discussed the inclusion and exclusion criteria, pre- and post-treatment evaluation, medications, goal setting, daily records, and notebook content. After discussion by the members for 6 months, the first edition of this notebook was published in 2014. The fifth edition of this notebook and the fourth edition of the manual for medical professionals were published in 2018 after several modifications.

### 2.5. Statistical Analysis

All statistical analyses were conducted using SPSS version 24 for Windows (IBM, Tokyo, Japan). The values for each subject were compared between pretreatment and post-treatment using the Wilcoxon rank sum test. Spearman’s rank correlation coefficient was used to analyze the correlation between the values of 2 measures. The significance level was set to less than 5%.

## 3. Results

### 3.1. Technical Note (Content and Use of the Ikiiki Rehabilitation Notebook)

This notebook was developed to encourage cooperation between physicians and medical staff (rehabilitation therapists, nurses, clinical psychologists, etc.) in treatment that incorporates the three components of exercise therapy, CBT, and patient education. “Ikiiki” means active in Japanese. To make this notebook as compact as possible and easy to use, the size was set to A5 (Figure 1), and the letters are large in consideration of the elderly. The medical staff carefully explains the contents of each job category in the table of contents (Figure 2). The first page clarifies the meaning of rehabilitation-related words, the significance of exercise therapy, and the cognitive and behavioral aims of the notebook (Figure 3). The next page explains how to use the notebook and provides guidance on writing the entry page for the day. The content regarding rehabilitation is explained in an easy-to-understand manner, with examples of rehabilitation exercises for low back (Figure 4), knee, and shoulder pain, which are frequently affected areas, together with photos and pictures of people performing stretching exercises, strength training, and whole-body exercises.

CBT is explained in an easy-to-understand language with examples (Figure 5). Chronic pain sufferers often lose their self-efficacy due to prolonged pain, and thus it is important to set clear goals. In rehabilitation medicine, it is standard to set goals with the patient, such as ambulation with a cane; perform ADL independently, except for bathing; and return to work. Accomplishing the set goals is the endpoint of this rehabilitation approach. In general, it is a desirable goal for chronic pain patients to improve their ADL and/or QOL, even if the pain intensity is not decreased. In this notebook, patients clarify and record their own unique goals. Patients set the goals by themselves to improve their ADL and/or QOL with support from the physician and medical staff (physical therapists) involved in this study. Patients are asked to set approximately three goals for motivation, in consideration of the patient’s aims. A goal of 6 months to 1 year is considered a long-term goal, and 1 month a short-term goal. It is desirable for the goals to be able to do something enjoyable that brings happiness, and to be able to objectively confirm their achievement. To promote a sense of accomplishment, it is also important to select a goal that is not too lofty and to create a content that is easy to achieve. Examples include: (a) return to work, return to school; (b) independence of daily life; (c) increase walking distance; (d) executing household chores; (e) going shopping alone; (f) resuming and expanding hobbies (engage in the art form of bonsai, attend a weekly floral arts class, etc.).

Patients were asked to write their feelings and thoughts in the notebooks daily to once a week, as well as to record their daily behavior and exercise routine (muscle exertion, gait distance or count of steps) (Figure 6). In addition, writing a “message to thank oneself for the effort” was encouraged to improve their self-efficacy. Frequency of recording in the notebook varied from daily to once per week, depending on the patients’ desire and feasibility. The medical staff did not force writing on the patients to avoid inducing psychological stress related to the therapy. Once every 2 weeks, the patients returned to the clinic to review the notebook with the medical staff. The contents of the notebook should be reviewed at each outpatient visit. If any improvement is made, even if trivial, the patient should be praised by physicians and medical staff. In cases of failure to achieve any improvement, the patients should be listened to and sympathized with regarding the cause and distress of the failure. Small achievements regarding patient effort and courage should be praised. The daily description should be “physical condition”, without using the word “pain”, and the medical staff should avoid talking about pain. It is important to listen to and sympathize with the patient, however, if the patient complains of pain (Figure 6).

On the daily entry page, patients are asked to write notes about their daily activities and the status of performing the prescribed exercise within a reasonable range. This is expected to provide a small sense of achievement and accumulation of effort every day, in addition to the analgesic effects of exercise. Separately entering daily “thought” and “feeling” helps the patients to objectively view themselves and makes it easier to correct their nonfunctional thinking (Figure 6). The purpose of the “message to thank oneself for the effort” is to promote self-care and self-efficacy by sending encouragement to oneself. Medical staff should praise the patient’s efforts and supportively increase the patient’s self-efficacy. The contents of the notebook should be checked regularly by the medical staff to complete the “Ikiiki level CHECK”, and feedback, such as praising adaptive behaviors and ideas, should be provided (performing positive reinforcement in learning theory). Together, these activities are designed to encourage behavioral change (Figure 6).

The patients were asked to reflect on their achievements and short-comings relating to their short-term and long-term goals at 1 month. If a patient’s physical performance improves, a new goal should be set, even a month later. Patients with chronic pain are likely to have endured it for a long time, preventing them from reaching their goals, so medical staff members should explain that treatment will take time and encourage them to continue with long-term outpatient support. To encourage the patient to be positive, medical staff members must focus on celebrating the achievements, no matter how small, and to not become critical. Mainly, patients themselves complete the notebook. Physicians and medical staff such as physical therapists can also write brief comments in the space of the notebook near the patient’s comments.

The final goals are to achieve the long-term goal at 6 months to 1 year, establish self-reliance, and to not be dependent on medical care (drugs and medical professionals). At the end of the outpatient clinic, the points of reflection and future matters are described on the reflection entry page, and the use of the rehabilitation notes is terminated. It is important that the outpatient clinic does not end suddenly, but that the frequency gradually decreases.

A 90-minute formal training seminar for this method, covering the exercise protocol and psychological approaches (including CBT), has been performed annually since 2014. The medical staff (one medical doctor who is a director as well as a lecturer, and two physical therapists) supervising the treatment in the present study have participated in coaching this seminar.

### 3.2. Outcomes and Changes in Each Parameter

To comply with the exclusion criteria, four patients underwent psychological consultation during the treatment to rule out severe psychological diseases, such as depression and schizophrenia. The outcomes were as follows: improvement in presenteeism (presenteeism refers to impaired productivity such as spending too much time at work or coming into work even when ill—actions that can negatively affect performance): nine cases; enhanced participation in hobbies: seven cases; improved school attendance: two cases; return to work: one case; improvement of self-care and/or self-efficacy: three cases; dropout because of patient desire: one case; dropout because of experiencing aggravated symptoms: one case (Table 1). In terms of the follow-up rate and follow-up periods, two cases dropped out, leaving 22 cases. Finally, the follow-up rate was 92% and follow-up duration for the 22 cases was 11.2 ± 4.7 months. The follow-up rate was high because this treatment was carefully introduced with detailed and repeated explanation, and patients were allowed a 1- to 2-week trial using the notebook before deciding whether to participate in the study. The post-treatment NRS, PDAS, HADS, PCS, AIS, PSEQ, and EQ-5D scores indicated significant improvement over the pretreatment scores (Table 2). The EQ-5D score at post-treatment significantly correlated with the pretreatment NRS (r = −0.45, *p* < 0.05, Figure 7a) and PCS (magnification) (r = −0.44, *p* < 0.05) scores. (Figure 7b).

## 4. Discussion

### 4.1. Psychological Factors following Exercise Treatment in Chronic Pain

Patients with chronic pain suffer from kinesiophobia, a fear that physical activity may increase pain, as an explanation of the fear-avoidance model [18]. Inactivity often results, and a vicious cycle of pain is likely to occur. A report on psychological factors related to a poor prognosis after exercise therapy for chronic low back pain, in which exercise therapy was performed for 4 weeks and the evaluation made 6 months later, demonstrated that the non-improved group had stronger fear avoidance, kinesiophobia, and depressive symptoms than the improved group [19]. This outcome suggests that a psychological element of the recovery should include a measure of fear-avoidance or kinesiophobia [19]. In the present study, the authors included the PCS, and demonstrated significant improvement from high scores (PCS at pretreatment: 34.95 ± 8.75) to normal scores on average (PCS at post-treatment: 24.00 ± 13.07) compared with the cutoff score (30 points).

### 4.2. Combined Effect of Exercise and CBT in Chronic Pain Treatment

Previous studies demonstrated that combining exercise and CBT significantly improves the intensity of pain, degree of dysfunction, catastrophizing thinking, and kinesiophobia with a moderate effect size at 1 month after the end of the treatment, compared with exercise only [20], with the effect continuing for at least 1 year [21]. General exercises in combination with CBT in the management of chronic low back pain are clinically more effective than general exercises alone [17]. Furthermore, the combined use of exercise, self-management education, and CBT reduces pain and catastrophizing, and prevents a decline in ADL ability [22]. Recent systematic reviews reported greater effectiveness of multidisciplinary rehabilitation with better patient education than exercise therapy alone [23].

A report from a Japanese multidisciplinary pain center showed significant improvement in a number of measures (e.g., pain intensity, disability, catastrophizing thoughts) after treatment for patients with chronic pain in group programs with physical exercise and CBT (weekly 2 h sessions for 9 weeks) [8]. In the present study, the PCS, which is commonly used to characterize chronic pain patients, showed significant improvement from pretreatment (total PCS score: 34.95) to post-treatment (24.00), compared with findings from a previous Japanese study (pretreatment total PCS score: 32.2 to post treatment total PCS score: 25.7) in which combination treatment with exercise and CBT was evaluated in patients with similar pain severity (pain VAS or NRS at pretreatment) [8]. As described in the “Technical Note (content and use of the Ikiiki Rehabilitation Notebook)” of the outcome, this notebook includes spaces to record thoughts, feelings, and messages to thank oneself for the effort to relieve negative thoughts and feelings, and to improve self-efficacy. These findings may demonstrate the usefulness of the Ikiiki notebook to fill these spaces as a CBT approach.

Only a few studies have reported the effects of combining CBT and exercise therapy in Japan, but Hirase et al. conducted a study targeting community-dwelling elderly people [16]. The patients of the present study had more severe conditions in terms of pain intensity, PCS, and PDAS than those in Hirase’s report before treatment. The NRS, PCS, and PDAS scores, however, improved to the same level in both studies after treatment. Although it is difficult to make a direct comparison because of the differences in the subject age, duration of treatment, and exercise prescription, it is noteworthy that use of the Ikiiki Rehabilitation Notebook method resulted in similar improvements as reported by Hirase et al. [16]. The present study is the first to use the “Ikiiki Rehabilitation Notebook” to educate individual patients about the pathogenesis of chronic pain and importance of exercise, and also to encourage written expression of thoughts, feelings, and gratitude to oneself for the effort put forth to relieve negative feelings and improve self-efficacy. These findings together indicate that the exercise facilitation method in combination with CBT using the “Ikiiki Rehabilitation Notebook” has the potential to significantly improve patients’ QOL with intractable chronic pain to the level of community-dwelling elderly people.

### 4.3. Catastrophizing Thoughts on the Outcome of QOL in Pain Treatment

Several studies have investigated the relationship between catastrophizing and both pain intensity and degree of disability [24,25]. Catastrophizing has a greater impact on disability than actual physical function [26]. Inoue and coworkers [8] reported a significant improvement in scores on a pain test that assessed understanding of lectures and awareness of pain and found that learning coping skills and how to confront pain reduces catastrophizing thoughts, disabilities, anxiety, and depression. In the present study, the relationship between PCS (magnification) at pretreatment and EQ-5D at post-treatment was negatively correlated (Figure 7b), indicating that patients with high PCS (magnification) before treatment tend to have a low QOL, even after using the Ikiiki Rehabilitation Notebook. In another dataset of the present study, pretreatment NRS and post-treatment EQ-5D were also negatively correlated, indicating that this finding is consistent with the outcome reported by George and co-workers [19] that pain intensity is predictive of 6-month recovery status after treating exercise.

## 5. Limitations of the Study

The present study has some limitations. First, there was no adequate control group; thus, it is unclear if using the Ikiiki Rehabilitation Notebook was better than exercise or CBT alone. This study included a small number of participants and a non-systematic approach to utilizing the notebook. The main purpose of this research, however, was to provide a detailed technical explanation of the method and to evaluate its effectiveness in a pilot study for patients with intractable pain. The outcome of the present study is consistent with data of previous Japanese studies [8,16]. A prospective randomized control trial is currently being conducted in five medical facilities to compare the data between combined treatment with exercise and CBT, and exercise alone.

Second, the treatment period varied from 3 to 22 months because the goal in this method differs from that of other methods in which a routine protocol is completed [8,16,17,22]. In the present method, the goal is to accomplish a long-term objective, establish self-reliance, and to decrease dependency on medical care, such as drugs and medical professionals. 

## 6. Conclusions

In this study, we aimed to describe a novel exercise facilitation method combined with CBT using the “Ikiiki Rehabilitation Notebook” for patients with intractable chronic pain, and to evaluate the effectiveness of using this method for patients. In 22 cases with chronic pain treated using this notebook, the pain intensity, ADL, catastrophizing thoughts, insomnia, self-efficacy and QOL were significantly improved after treatment. It is unclear if the score changes are clinically meaningful because the treatment results may reflect several types of therapeutic interventions using this notebook. Further investigation in a prospective study, such as a randomized controlled trial, is needed to evaluate the effectiveness of using this notebook.

## Figures and Tables

**Figure 1 healthcare-09-01209-f001:**
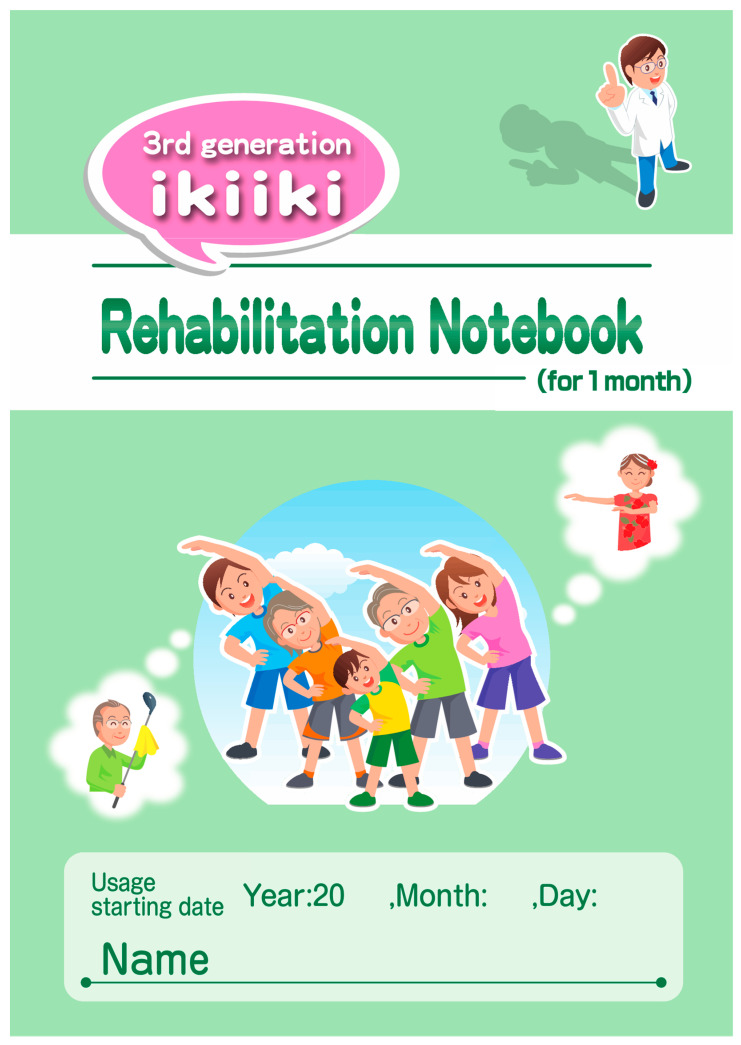
Cover sheet of the Ikiiki Rehabilitation Notebook.

**Figure 2 healthcare-09-01209-f002:**
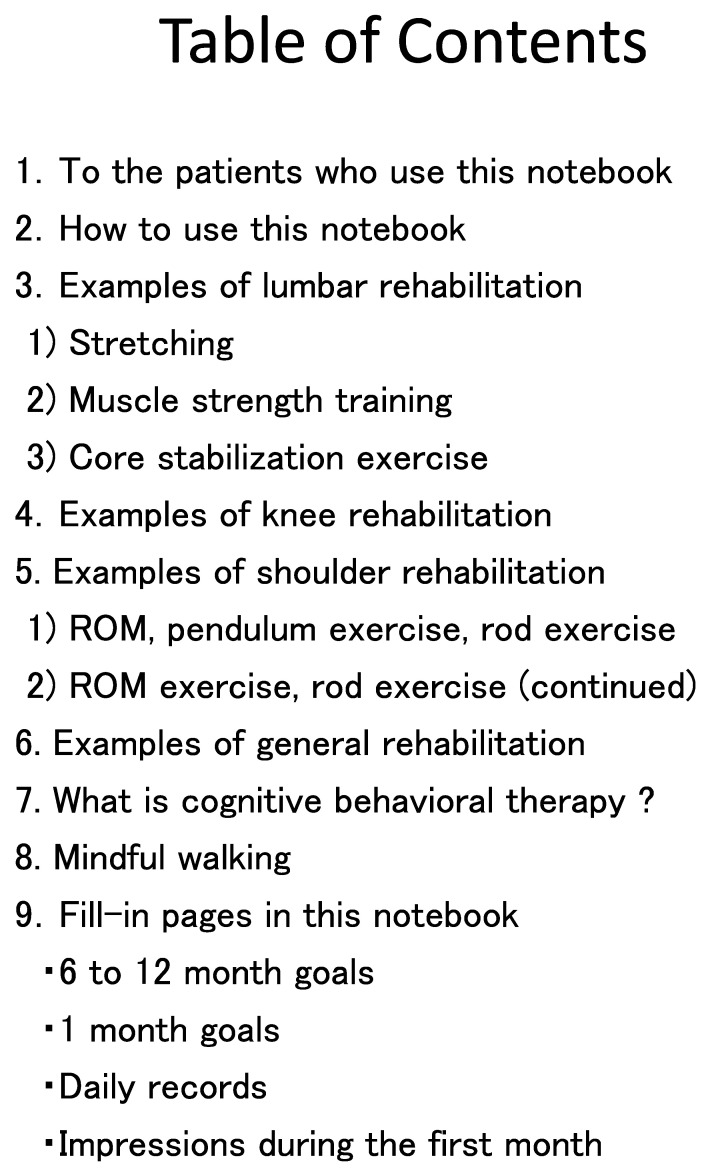
Table of contents in the Ikiiki Rehabilitation Notebook.

**Figure 3 healthcare-09-01209-f003:**
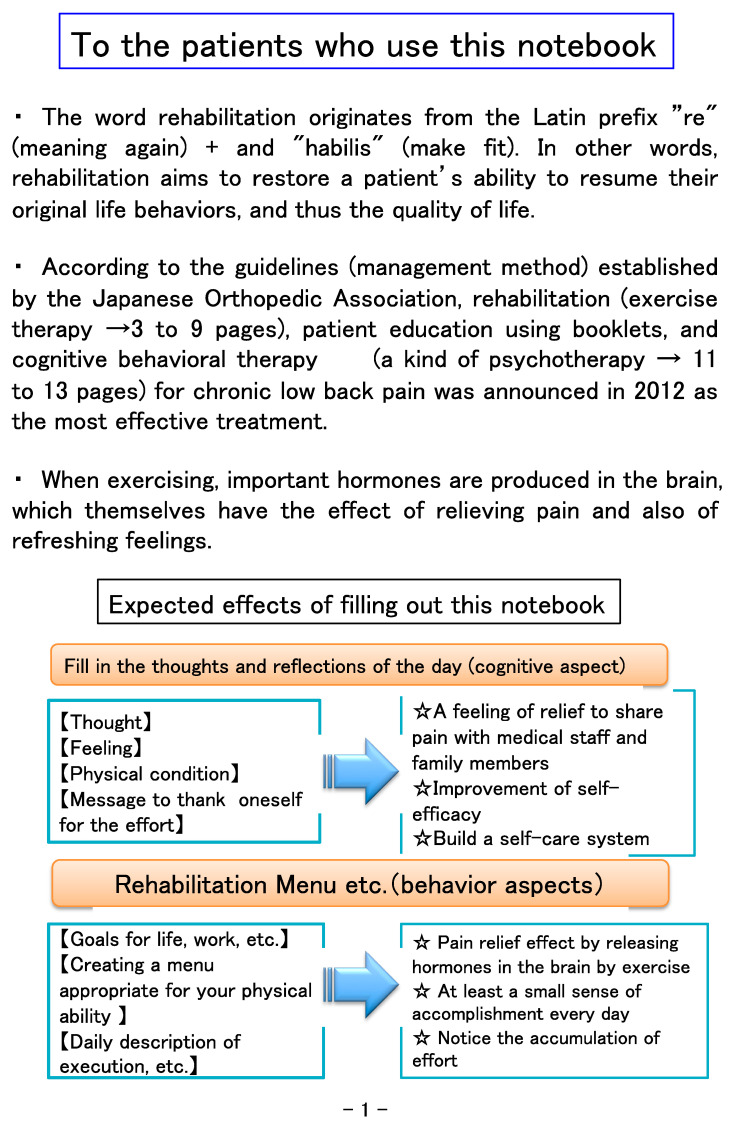
Objectives for using this notebook.

**Figure 4 healthcare-09-01209-f004:**
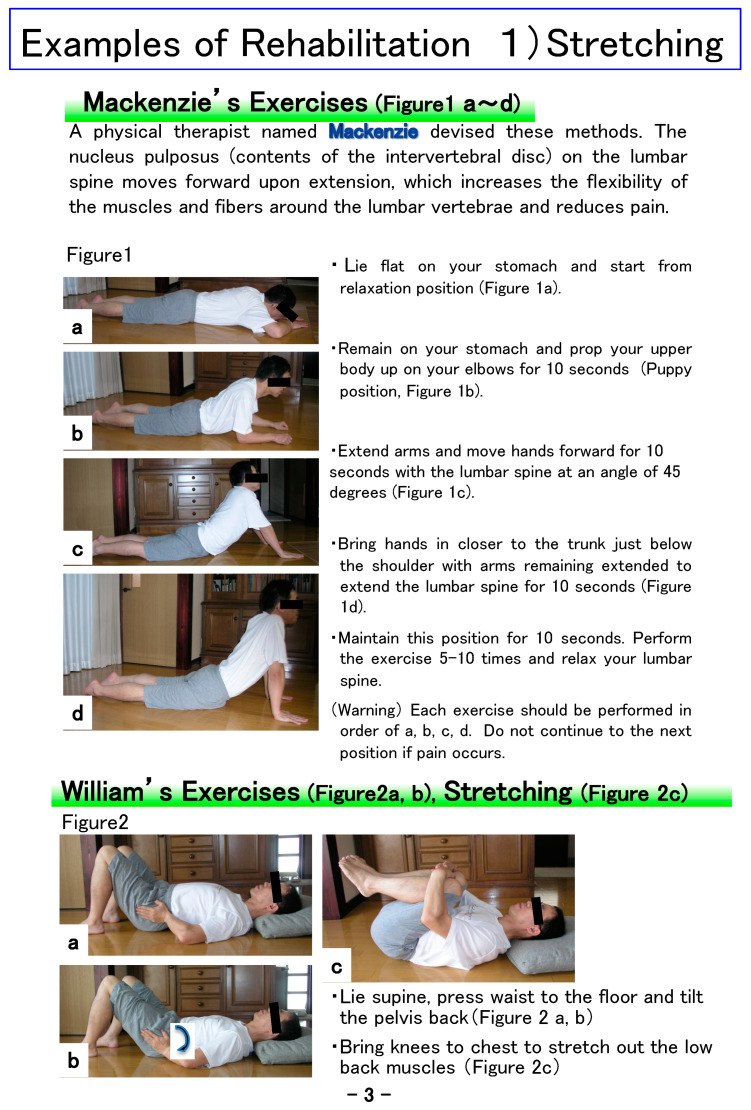
Examples of lumbar spine rehabilitation.

**Figure 5 healthcare-09-01209-f005:**
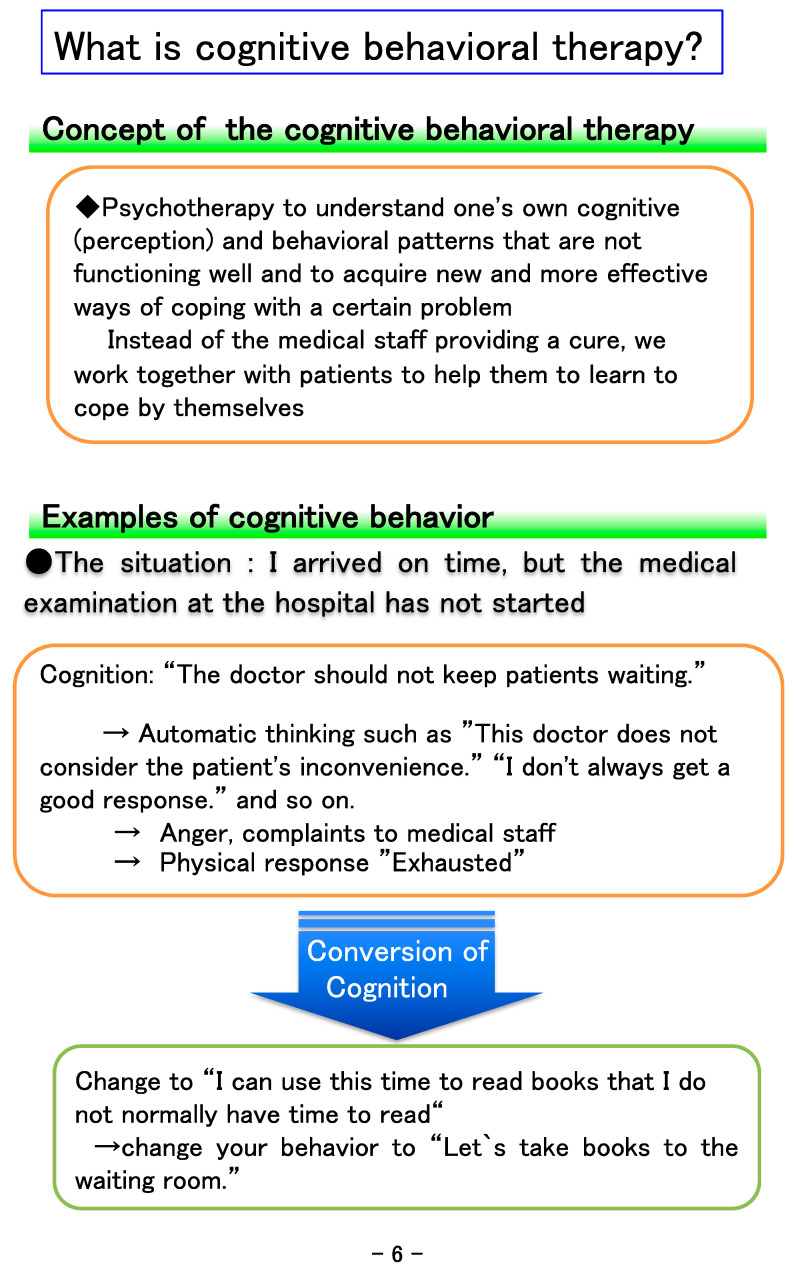
Explanation of cognitive behavioral therapy.

**Figure 6 healthcare-09-01209-f006:**
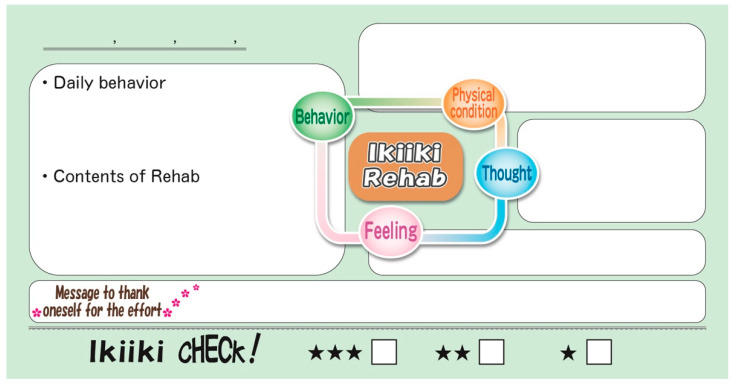
Daily entry page in the Ikiiki Rehabilitation Notebook.

**Figure 7 healthcare-09-01209-f007:**
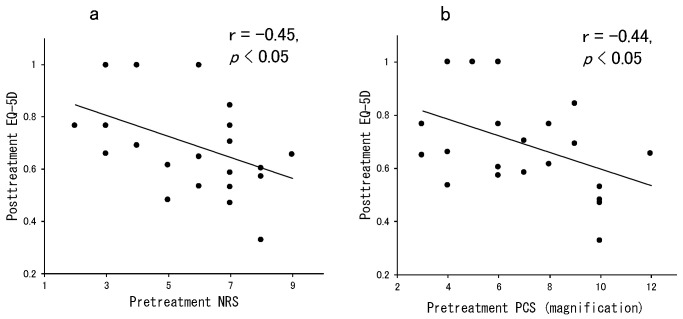
Relationship between NRS or PCS (magnification) score at pretreatment and EQ-5D score post-treatment. Notes: The NRS score at pretreatment was significantly correlated with the EQ-5D score post-treatment ((**a**), r = −0.45, *p* < 0.05), and the PCS (magnification) score at pretreatment was significantly correlated with the EQ-5D score post-treatment ((**b**), r = −0.44, *p* < 0.05).

**Table 1 healthcare-09-01209-t001:** Characteristics of 24 cases.

Case No.	Age	Sex	Period of Pain Continuity (Month)	Period of Treatment (Month)	Number of Medical Facilities *	Clinical Diagnosis	Outcome for Goal
1	39	male	58	19	3	chronic lower extremity pain	presenteeism↑
2	73	female	36	12	2	lumbar spondylosis	hobbies↑
3	30	female	5	11	1	chronic low back pain	presenteeism↑
4	20	female	36	9	5	chronic knee pain	school↑
5	72	female	7	6	2	osteoarthritis of the knee	self-care↑
6	30	male	120	3	2	chronic low back pain	presenteeism↑
7	62	female	76	10	3	chronic low back pain	self-care↑
8	77	male	96	6	6	lumbar spondylosis	hobbies↑
9	19	female	17	13	3	chronic lower extremity pain	school↑
10	40	male	24	14	3	chronic low back pain	presenteeism↑
11	35	male	200	16	6	chronic low back pain	presenteeism↑
12	63	female	62	9	3	chronic low back pain	self-care↑
13	46	female	33	9	1	chronic lower extremity pain	presenteeism↑
14	41	female	32	12	9	chronic neck pain	return to work
15	73	female	26	11	2	lumbar spondylosis	hobbies↑
16	40	female	93	9	3	chronic low back pain	hobbies↑
17	68	female	66	6	6	chronic low back pain	hobbies↑
18	72	female	94	7	5	lumbar spondylosis	hobbies↑
19	46	female	48	22	3	chronic low back pain	presenteeism↑
20	47	female	38	14	5	chronic knee pain	presenteeism↑
21	66	male	24	10	2	lumbar spondylosis	hobbies↑
22	73	male	83	18	4	chronic plantar pain	presenteeism↑
23	36	male	84	6	5	chronic low back pain	drop-out because of patient’s desire
24	52	female	33	6	7	chronic neck pain	drop-out because of experiencing aggravated symptoms
AVG	51.5		58.0	10.8	3.8		
SD	18.0		43.5	4.7	2.0		

Notes: presenteeism↑: improvement of presenteeism, hobbies↑: enhanced participation in hobbies, school↑: improved school attendance, self-care↑: improvement of self-care and/or self-efficacy, *: number of medical facilities treated before 1st visit at the Niigata University Hospital, AVG: average, SD: standard deviation.

**Table 2 healthcare-09-01209-t002:** Pain-related assessments at pretreatment and post-treatment.

Pain-Related Assessments	Pretreatment	Post-Treatment	*p*-Value
NRS	5.8 ± 1.9	4.0 ± 2.0	0.0004
PDAS	25.95 ± 11.54	16.73 ± 11.04	0.0002
HADS Anxiety	8.09 ± 3.54	4.95 ± 3.62	0.003
HADS Depression	7.73 ± 4.38	4.77 ± 3.69	0.027
PCS Total	34.95 ± 8.75	24.00 ± 13.07	0.0006
PCS rumination	13.45 ± 2.60	9.50 ± 4.71	0.0005
PCS magnification	6.95 ± 2.57	4.73 ± 3.48	0.0017
PCS helplessness	14.55 ± 4.69	9.50 ± 5.63	0.0007
AIS	9.05 ± 5.13	5.59 ± 4.14	0.0103
PSEQ	30.45 ± 15.66	39.19 ± 12.35	0.0152
EQ-5D	0.55 ± 0.11	0.69 ± 0.19	0.0016

Notes: Data are presented as mean ± SD (standard deviation) unless otherwise indicated. Abbreviations: NRS, Numerical Rating Scale; PDAS, Pain Disability Assessment Scale; HADS, Hospital Anxiety and Depression Scale; PCS, Pain Catastrophizing Scale; AIS, Athens Insomnia Scale; PSEQ, Pain Self-Efficacy Questionnaire; EQ-5D, EuroQol 5 Dimension.

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
