# Peer review of "A Novel Exercise Facilitation Method in Combination with Cognitive Behavioral Therapy Using the Ikiiki Rehabilitation Notebook for Intractable Chronic Pain: Technical Report and 22 Cases"

_healthcare, 2021, doi:10.3390/healthcare9091209_

Round 1
Reviewer 1 Report
Comments and Suggestions for Authors
Manuscript entitled “A novel exercise facilitation method in combination with cognitive behavioral therapy using the Ikiiki Rehabilitation Note book for intractable chronic pain: technical report and 22 cases” addresses the treatment effectiveness for the patients with chronic pain. The manuscript is mainly focused on CBT. This is an important topic. However, some information needs to be clarified and rewritten.
In general, 2.Materials and Methods; 2.1.Patient information are very superficial and does not bring the patients conditions on the details because several diseases are mixed. I recommend the authors show the data in table etc. on the details about patient’s background data in each pain.
In addition, exclusion and inclusion criteria are unclear. Please reveal that how to diagnose the psychologic diseases for exclusion. I recommend you to show all of the patient’s data using the notebook, in addition, how many patients could able to continue and complete the notebook. If possible, the authors show follow-up rate and follow-up periods. In this paper, it sounds that the authors only selected the good results patients.
Most important point is that only using this notebook doesn’t lead the CBT treatment.
The author needs to show this technical note are really lead the patients to CBT if you focus on the CBT.
This note is one of the important treatment for patients with chronic pain to encourage the self-exercise. However, the readers doesn’t realize what factors are effective on the patients used the notebook only from the results.
If the medical staff used the notebook following the sophisticated CBT training, please show the one.
In the results, the authors showed the changes in each parameter and mentioned that the treatment effect by the notebook. It was the treatment results following many kinds of therapeutic interventions with the notebook. In addition, there were no control group. I think these score changes are still unknown whether it was clinically meaningful changes or not.
I think this paper is only technical note, so doesn’t mentioned about CBT because the authors have to show this one is really lead to CBT treatment scientifically to show the patients data. Only mention about the possibility is better way.
In addition, the results cannot lead the treatment effect, so only show in each case result.
I think showing all the data are no issues, however, the data interpretation are needed to be more careful.
Reviewer 2 Report
Physiotherapy for spine problems is patient centered; your approach, based only on computer training, is not an EBM - centered treatment. Patients improvements could not be referred to your protocol but to numerous bias sources.
Reviewer 3 Report
Reviewer report
I would like to thank the authors for undertaking this important study. Pain management integrating psychological and social support along with enhanced health care practitioner – patient interaction is most desirable. However, the manuscript requires extensive revision. My recommendations are listed below:
- Formatting/ Presentation
- Information presented in the results section belongs into the methods (e.g., lines 128 - 142
- The discussion is actually a literature review
- The discussion should compare the study’s findings with the literature (are findings corroborated or did you find inconsistencies and possible explanations)
- The limitation should be presented before the conclusion and include a statement that the findings cannot be generalised due to the low number of participants and non-systematic approach of utilising the notebook.
- The conclusion should be extended. It should restate the aim of the study, provide a brief summary of the main findings, followed by concluding remarks (what did the researchers conclude from this study), and an outlook what should be done next (e.g., further research is required).
- Academic writing
- At times clarity of expression should be improved – meanings of sentences are unclear and grammatical errors were observed. A thorough editing of the whole manuscript is required.
- Specific points
- You mentioned QOL but only provided the abbreviation – state “quality of life” in the abstract and again in the methods section
- I suggest you refrain from stating that this was an ‘evaluation’, which it clearly was not. Perhaps it is more accurate referring to a ‘case study’ using 22 participants of which the majority was female.
- Participants should not be referred to as ‘subjects’
- Did the researchers answer a specific research question? If so, this should be clearly stated and explained why this study was undertaken and for whom
- More clarity is required about who completed the notebook (only participants?) – mention was made that it was developed to improve collaboration between health professionals, which is confusing.
- I suggest that you place the notebook as an appendix
- It is important that you define terms (i.e., “self-efficacy” and “presenteism”)
- Clearly explain in the methods section what patients were required to do and provide further information on the frequency (daily, or once/several times per week) and what you mean by patients being asked to reflect upon.
- You mentioned “goals” but it was not clear what this meant and who set goals for what.
Round 2
Reviewer 1 Report
This revised article is well written and scientifically sounds good.
No need to rewrite additionally.
Author Response
I appreciate valuable Reviewer1' comment.
Reviewer 3 Report
Some amendments were made however the manuscript is below the expected standard.
Author Response
I appreciate Reviewer 3' comments. Academic Editor gave me the specific comments. I responsed to the comments as follows:
(Academic Editor's comment)Although revisions are partly carried out, several flaws need to be addressed.This paper describes a technical note, but it does not describe the unmet need and rationale that led to the development of the note as a methodology. Please have the method section revised and describe what theory was used to establish each part. (My comments)Thank you for your reasonable advice. Therefore, I added [2.4.] in the Method as follows:
2.4. Background and development of the Ikiiki Rehabilitation Notebook
In a study reported by Hirase et al. to evaluate a combined protocol of CBT and exercise therapy, physical therapists providing support for the community exercise classes checked the participant’s diaries once a week and advised participants to alter their pain awareness by focusing on increasing their daily activities [16]. The diary used in this study, however, did not include queries of psychosomatic content such as ”thoughts” and “feelings”, which is important to relieve the catastrophizing thoughts that are characteristic of chronic pain patients. To the best of our knowledge, there are no reports of studies in which participants used a notebook to record their thoughts, feelings, message to thank oneself for the effort toward combined CBT and exercise therapy [8], [17].
Therefore, the first author of the present study (a rehabilitation physician and board member of the Japanese Association for the study of Musculoskeletal Pain) assumed leadership of the study and recruited members from the boards of the same associations to develop the notebook used here. Members comprised 2 psychosomatic physicians, a psychiatrist, a physical therapist, 2 rehabilitation physicians, and a pain clinician. For the exercise therapy, the physical therapist, rehabilitation physicians, and pain clinician mainly discussed examples of short-term and long-term goals, exercise therapy types and methods, and methods for monitoring the number of steps taken per day. For CBT, the psychosomatic physicians and psychiatrist discussed methods of describing thoughts and feelings in daily records based on CBT theory. The idea of writing a "message to thank oneself for the effort", as the most distinctive feature of this notebook, was suggested by the psychosomatic physician (second author in the present study) from the perspective of improving the low self-efficacy that is considered a characteristic of chronic pain patients. Following submission of the springboard for discussion from each member, all members together discussed the inclusion and exclusion criteria, pre- and posttreatment evaluation, medications, goal setting, daily records, and notebook content. After discussion by the members for 6 months, the first edition of this notebook was published in 2014. The fifth edition of this notebook and the fourth edition of the manual for medical professionals were published in 2018 after several modifications.